# Adsorption of Pb$^{2+}$ Ions from Aqueous Solution onto Porous Kappa-Carrageenan/Cellulose Hydrogels: Isotherm and Kinetics Study

Karuppiah Kalaiselvi [1,*,†], Sonaimuthu Mohandoss [2,†], Naushad Ahmad [3,†], Mohammad Rizwan Khan [3] and Ranjith Kumar Manoharan [4,*]

1   Department of Chemistry, Government Arts and Science College, Paramakudi 623701, Tamil Nadu, India
2   School of Chemical Engineering, Yeungnam University, Gyeongsan 38541, Republic of Korea; drsmohandoss@yu.ac.kr
3   Department of Chemistry, College of Science, King Saud University, Riyadh 11451, Saudi Arabia; anaushad@ksu.edu.sa (N.A.); mrkhan@ksu.edu.sa (M.R.K.)
4   Department of Civil Engineering, Yeungnam University, Gyeongsan 38541, Republic of Korea
*   Correspondence: selvik495@gmail.com (K.K.); mrkumar@ynu.ac.kr (R.K.M.)
†   These authors contributed equally to this work.

**Abstract:** Heavy metal ion pollution poses severe health risks. In this study, a kappa-carrageenan/cellulose (κ-CG/CL) hydrogel was prepared using a facile one-step method to remove Pb$^{2+}$ ions from aqueous solutions. The functional groups and crystallinity nature of κ-CG/CL hydrogel have been identified via Fourier-transform infrared spectroscopy (FTIR) and X-ray diffraction (XRD). In contrast, the porous morphology and size distribution on the surface of κ-CG/CL hydrogel with a pore size of 1–10 μm were identified using scanning electron microscope (SEM) and Brunauer–Emmett–Teller (BET) surface area analysis. The as-prepared κ-CG/CL hydrogel effectively removed Pb$^{2+}$ ions, primary environmental pollutants. The effects of pH and contact time on Pb$^{2+}$ adsorption were studied along with the adsorption isotherms and kinetics of Pb$^{2+}$ adsorption onto the hydrogels from aqueous solutions. Notably, the aqueous solutions were effectively treated with the prepared κ-CG/CL hydrogels to remove Pb$^{2+}$ ions. The adsorption results fit well with pseudo-first- and second-order kinetic, Elovich, intra-particle diffusion, and Langmuir and Freundlich isotherm models. Based on the fitting results, the maximum adsorption capacity was obtained with the Freundlich isotherm model of κ-CG/CL hydrogel found to be 486 ± 28.5 mg/g (79%). Reusability studies revealed that the κ-CG/CL hydrogel could remove Pb$^{2+}$ ions with more than 79% removal efficiency after eight adsorption–desorption cycles. In addition, its mechanism for efficiently adsorbing and removal of Pb$^{2+}$ ions was analyzed. These findings imply that the κ-CG/CL hydrogel has substantial potential for application in removing and recycling heavy metal ions from aqueous solutions.

**Keywords:** adsorption; Pb$^{2+}$; hydrogels; kinetics; isotherm

## 1. Introduction

Water pollution due to heavy metals is a critical concern [1]. A small amount of heavy metal in wastewater poses a serious threat to natural ecosystems and human health because heavy metal ions can remain in water bodies for a long time and be enriched by the food chain [2–4]. Heavy metals in contaminated water can be removed using various techniques. Adsorption is an attractive alternative to conventional treatments, such as precipitation, ion exchange, and coagulation–flocculation [5,6]. Research has focused on developing high-performance, easy-to-prepare, and low-cost adsorbents in recent years [7–11]. There has been a great deal of attention in recent years focused on removing heavy metals from water systems. Specific metal ions present in the environment, including Cd$^{2+}$, Ni$^{2+}$, Pb$^{2+}$, and Hg$^{2+}$, which have carcinogenic and toxic properties even

at extremely low concentrations, pose a severe threat to human health [12]. The use of polymeric materials has become increasingly popular to remove and separate heavy metal ions due to environmental pollutants resulting from industries' waste products. $Pb^{2+}$ ions are prevalent in nature but are considered dangerous because of their carcinogenic properties [13]. Several recent studies on Pb toxicity indicate that Pb exposure affects the nervous and reproductive systems and the kidneys, liver, and brain [14]. Owing to their potential risks, $Pb^{2+}$ ions have been confirmed as a leading cause of public health anxiety by the World Health Organization (WHO) [12]. Environmental regulations have increased the importance of treating $Pb^{2+}$-contaminated water. In addition to offering operational flexibility, convenience, and economic benefits, adsorption is a promising method for removing $Pb^{2+}$ from water [15].

Three-dimensional hydrogels contain large amounts of water. Biopolymers are frequently cross-linked chemically or physically to form network gels. In recent years, polymeric hydrogels have been considered adsorbents for removing toxic metal ions [16]. In addition to the ability to incorporate chelating groups into polymeric networks, the polymers exhibit a large surface area, high adsorption capacity, and chemical stability [17]. Various natural polymers are currently used in low-cost water purification technologies to remove metal ions from water. These polymers include cellulose (CL), starch, dextran, kappa-carrageenan (κ-CG), and chitosan [18]. Hydrogels absorb water according to their functional group, state of water, and cross-linking network density. Furthermore, they exhibit osmotic pressure and possess hydrophilic hydroxyl (-OH), amide ($-NH_2$), carboxyl (-COOH), and sulfonic acid ($-SO_3H$) groups in their polymer networks [19]. For the removal of toxic metal ions, these functional groups can be utilized in hydrogel networks. Cellulose (CL) is one of the most abundant, environmentally friendly, renewable, and biodegradable polymers on earth, suggesting its potential as a green adsorbent resource [20]. The inherent crystallinity of CL gives it low density, high strength, and high stiffness. It is a biocompatible, biodegradable, renewable, mechanically strong, eco-friendly, and non-toxic material [21]. Environmentally friendly and biocompatible products are in high demand, and it is regarded as an almost inexhaustible source of raw materials. CL is a polysaccharide comprising linear chains of linked D-glucose units and an essential source of food and energy [22]. Hydrogels with various structures and properties can be prepared using CL because it contains abundant hydroxyl groups [23]. One advantage of such hydrogels is that they can serve as platforms.

Recently, nanomaterials have been widely used as adsorbents because of their highly specific surfaces and active sites. Owing to their high adsorption capacities, they are effective in removing pollutants. Kappa-carrageenan (κ-CG) is a non-toxic sulfated polysaccharide with a structure made up of α(1→4) D-galactose-4-sulphate and β(1→3) 3,6-anhydro-D-galactose, which is extracted from specific species of red seaweed [24]. Gel formation is mainly achieved using iota (ι)- and κ-CG. In the structure of κ-CG, hydroxyls and sulfate groups make it hydrophilic. The chemical structure of κ-CG is slightly acidic or neutral in accordance with its natural abundance. κ-CG hydrogel can reduce or eliminate toxicity in biomedical applications. Moreover, κ-CG serves as a backbone for the synthesis of a hydrogel with CL likely to improve the capacity of the newly developed adsorbents, which was used to remove $Pb^{2+}$ ions from an aqueous solution [25–27].

In the past decade, CL and modified CL has been reported to be effective in removing heavy metal ions, such as $Pb^{2+}$, $Hg^{2+}$, $Cu^{2+}$, $Ni^{2+}$, $Zn^{2+}$, $Fe^{3+}$, $Cd^{2+}$, and $Cs^+$ from aqueous solutions [28–33]. Zhou et al. used modified CL for the removal of $Cd^{2+}$, $Hg^{2+}$, and acid fuchsin from aqueous solution [28]. Cho et al. prepared microcrystalline CL-based porous material for heavy metal removal from an aqueous solution [29]. Liu et al. used amide-functionalized CL-based adsorbents for the heavy metal ions and anionic dyes from aqueous solutions [30]. Moreover, pure CL-based hydrogels used the adsorption behavior of heavy metal ions from aqueous solutions [31–33]. Therefore, experiments show CL is a practical material for modifying κ-CG, which improves the adsorption capacity of $Pb^{2+}$ ions from aqueous solutions. Hence, based on the survey of the literature, this work aims to

evaluate κ-CG and CL in the form of κ-CG/CL hydrogels for the removal of $Pb^{2+}$ ions from aqueous solutions. To the best of the author's knowledge, there is no analysis reported in the literature on the removal of $Pb^{2+}$ ions by using κ-CG/CL hydrogel.

The objective of the present work was to investigate a κ-CG/CL hydrogel based on kappa-carrageenan (κ-CG) and cellulose (CL) as a sorbent for the removal of $Pb^{2+}$ ions from aqueous solutions. The novelty of this work is that it identifies various significant parameters for the removal of $Pb^{2+}$ metal ions from its aqueous solution using κ-CG hydrogels. The structures of κ-CG, CL, and freeze-dried κ-CG/CL hydrogel were characterized using FTIR and XRD analysis. The surface morphology and surface area of κ-CG/CL hydrogels samples were performed by SEM and BET analysis. The influence of pH, dosage, time, concentration, and regeneration properties was also investigated. Furthermore, we analyzed the kinetics and adsorption mechanism of $Pb^{2+}$ ions using pseudo-first- and second-order kinetic, Elovich, and intra-particle diffusion models. We evaluated the equilibrium adsorption data using Langmuir and Freundlich isotherms. Eight successive adsorption–desorption cycles were used to assess the desorption efficiency of the adsorbents and their reusability.

## 2. Materials and Methods

### 2.1. Materials

Kappa-carrageenan (κ-CG) and microcrystalline cellulose (CL) were purchased from Merck (Seoul, Republic of Korea). Lead nitrate ($Pb(NO_3)_2$) and calcium carbonate ($CaCO_3$) were obtained from TCI Chemical Company, Seoul, Republic of Korea. All chemicals used in the experiments were of analytical grade, and MilliQ Biocel A10, Millipore, Burlington, MA, USA, deionized water was used. Metal salts were weighed and transferred into a volumetric flask of 100 mL and prepared as stock solutions. A solution of deionized water was added to dissolve the metals completely. To achieve analytical accuracy and precision, it is crucial to use reagents and standards of the highest purity when calibrating the atomic absorption spectrometry (AAS) instrument.

### 2.2. Preparation of Hydrogels

κ-CG/CL-based hydrogels were formulated using around 70:30 κ-CG: CL ratio with 1.0% $CaCO_3$. Briefly, for a typical hydrogel synthesis, 0.50 g of κ-CG and 0.20 g CL were dissolved in 15 mL of distilled water at 70 °C before mixing with 0.20 g $CaCO_3$ dissolved in 20 mL of distilled water. A transparent, viscous, and homogeneous solution was obtained after stirring for 1 h. We poured the mixed solution into a glass mold made in the laboratory and incubated it for two days at 5 °C to produce sheet-shaped gels. κ-CG/CL hydrogel samples were at equilibrium with room temperature for 24 h before drying at 37 °C in an oven which was left overnight.

### 2.3. Characterizations

The FTIR spectra of pure molecules and κ-CG/CL hydrogel were measured using a spectrophotometer (Perkin Elmer) ranging from 4000 to 400 $cm^{-1}$. The powder XRD analysis of pure molecules and κ-CG/CL was carried out using a Philips/PANalytical X'Pert MRD in a 2θ from 10 to 80° using Cu Kα (λ = 0.1540 nm). The SEM images of the κ-CG/CL hydrogel were obtained using a scanning electron microscope (Leo Supra 50VP, Jena, Germany) at 2 keV. The BET-specific surface area of the κ-CG/CL hydrogel was measured using a (BEL, Japan Inc., Osaka, Japan) Belsorp II-mini.

### 2.4. Adsorption Experiments

Adsorption experiments of $Pb^{2+}$ ions on the κ-CG/CL hydrogel were performed at 27 °C by introducing 30 mg dry κ-CG/CL hydrogel to 100 mL conical flasks containing 50 mL $Pb^{2+}$ ions in an aqueous solution. A constant-temperature oscillator was used to agitate the conical flasks at 300 RPM. Suitable amounts of NaOH and HCl were added to the above solution to adjust the pH. We investigated the effect of pH values ranging

from 1.0 to 6.0 on metal adsorption. During the adsorption kinetics experiments, the contact time varied from 30 to 48 min. The adsorption isotherm experiments used metal ion concentrations of 50–600 mg/L. $Pb^{2+}$ residual concentrations in bottles after adsorption were determined using atomic absorption spectroscopy (AAS). The following equations were calculated to compute the adsorption capacity equilibrium ($Q_m$, mg/g) and the quantity of $Pb^{2+}$ ions adsorbed time t ($Q_t$, mg/g):

$$Q_t = \frac{(C_0 - C_t)V}{W} \tag{1}$$

$$Q_m = \frac{(C_0 - C_e)V}{W} \tag{2}$$

In the above equations, $Q_t$ (mg.g$^{-1}$) denotes the quantity adsorbed at a specific time, $Q_m$ (mg/g$^{-1}$) represents the quantity adsorbed at equilibrium, $C_o$ represents the initial concentration, $C_e$ denotes the concentration of equilibrium, $C_t$ denotes the $Pb^{2+}$ ions concentration at a specific time (h), V (mL) represents the volume of the $Pb^{2+}$ ion, and W (mg) represents the mass of the dried κ-CG/CL hydrogel.

## 2.5. Desorption Experiments

The reusability of the κ-CG/CL hydrogel was examined through sequential adsorption–desorption cycles. To conduct desorption tests, known quantities of the κ-CG/CL hydrogel loaded with $Pb^{2+}$ ions were submerged in 30 mL of 0.1 N $HNO_3$ for 2 h at room temperature. The hydrogel was collected, washed, and reused, and the experiment was repeated. During the adsorption–desorption tests, no important loss in the removal efficiency was detected after eight cycles. After each adsorption cycle, AAS was used to determine the $Pb^{2+}$ ions concentration in the aqueous phase.

## 2.6. Adsorption Kinetics

When evaluating the adsorption efficiency of adsorbents, adsorption kinetics demonstration is a significant part of determining the equilibration and contact times. It is desirable to start measuring adsorption kinetics within one minute and to end measurements when the adsorption process has reached equilibrium. Therefore, the κ-CG/CL hydrogel adsorption capacities are investigated at an initial time of 30 s–1500 min under the baseline operating conditions. To investigate the kinetic behavior of the κ-CG/CL hydrogel in the removal of $Pb^{2+}$ ions from aqueous solutions, pseudo-first and second-order (Equations (3) and (4)) [34,35], Elovich (Equation (5)) [36], and intra-particle diffusion (Equation (6)) [37] models were fitted to the experimental data.

$$Q_t = Q_e \left(1 - e^{-k_1 t}\right) \tag{3}$$

$$Q_t = \frac{Q_e 2k_2 t}{1 + k_2 Q_e t} \tag{4}$$

where the rate constants for the pseudo-first- and second-order equations are $k_1$ (min$^{-1}$) and $k_2$ (g·(mg min)$^{-1}$), respectively. The equilibrium of adsorption capacity is $Q_e$ (mg/g), while the adsorption capacity at any time t (min) is $Q_t$ (mg/g).

$$Q_t = \frac{1}{\beta}\ln(1 + \alpha\beta t) \tag{5}$$

where $Q_t$ (mg/g) is the initial rate constant, $\alpha$ (mg(g min)$^{-1}$) is the adsorption capacity at any given time, and $\beta$ (mg/g) is the desorption constant during each trial.

$$Q_t = k_{t,1} t^{0.5} + C_1 \tag{6}$$

where $k_{t,1}$ (g (mg min$^{-0.5}$)$^{-1}$) is the rate constant of the intra-particle diffusion model and $C_1$ (mg/g) is a constant relating to the boundary layer thickness, where a greater value of $C_i$ correlates to a more significant effect on the constraining boundary layer.

### 2.7. Adsorption Isotherms

Adsorption isotherms are necessary to establish adsorption studies and understand the relationship between the adsorbent (κ-CG/CL hydrogel) and the adsorbate content in aqueous media (Pb$^{2+}$ ion solutions). In addition to using a single starting adsorbate concentration, researchers should be able to use several different Pb$^{2+}$ concentrations and temperatures, such as 298, 308, and 318 K, instead of a single starting concentration and temperature. The Langmuir model (Equation (7)), as well as the Freundlich model (Equation (8)), have been used as models for adsorption isotherms in the literature [38,39]. We utilized these two isotherm models to investigate the adsorption isotherms. Because of their practicability, simplicity, and ease of interpretation, the model parameters are readily applicable.

$$R_L = \frac{1}{1 + K_L C_e} \tag{7}$$

where $K_L$ is the Langmuir equilibrium constant, and $C_e$ (mg/L) is the initial adsorbate content. In addition, $R_L = 0$ signifies irreversible adsorption, and $R_L > 1$ denotes unfavorable adsorption.

$$Q_m = K_F C_e{}^n \tag{8}$$

The equilibrium adsorbate composition is denoted by $C_e$ (mg/L), the equilibrium adsorption capacity is represented by $Q_m$ (mg/g), the equilibrium Freundlich component is represented by $K_F$ (mg/g)/(mg/L)$^n$, and the equilibrium Freundlich intensity component is represented by n (dimensionless), which indicates the strength of the adsorption driving force or the surface heterogeneity.

### 2.8. Statistical Analysis

All statistical analyses were performed by using Origin 9.0 software. The results are expressed as the means ± standard deviations and analyzed by one-way analysis of variance. The statistical significance was determined by $p < 0.05$.

## 3. Results and Discussion
### 3.1. FTIR Spectra of κ-CG/CL Hydrogel

FTIR spectra of the kappa-carrageenan (κ-CG), cellulose (CL), and κ -CG/CL hydrogels are shown in Figure 1. As shown in Figure 1a,b, both κ-CG and CL exhibit common characteristic absorption broad peaks between 3500 and 3200 cm$^{-1}$, which can be attributed to the stretching vibration of –OH [40,41]. A peak between 2917 and 2903 cm$^{-1}$ was also observed. These substances consisted of -CH$_3$ and -CH$_2$ groups stretched in the -CH mode. In the κ-CG/CL hydrogel, the stretching vibration peak of -OH was constricted and relocated to a lower wavenumber at 3315 cm$^{-1}$, suggesting that κ-CG and CL may form hydrogen bonds. Moreover, in pure κ-CG and CL, distinctive peaks associated with the C-O and the band of a carboxyl group (−C=O) stretching vibration were detected at 1054/1023 cm$^{-1}$ and 1635 cm$^{-1}$, respectively. However, in the κ-CG/CL hydrogel, such a peak appeared at 1045 cm$^{-1}$. The peaks of -OH and C-O were narrower, shifted, and more intense than those of pure κ-CG and CL, as shown in Figure 1c, proving that the modification of κ-CG and CL was successful.

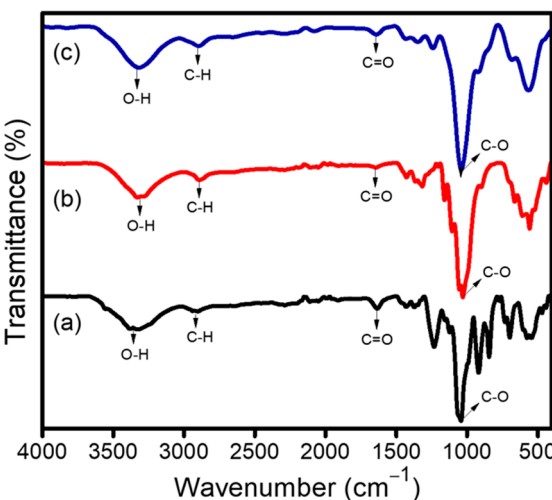

**Figure 1.** FTIR spectra of (a) kappa-carrageenan (κ-CG), (b) cellulose (CL), and (c) κ-CG/CL hydrogel.

### 3.2. XRD Pattern of κ-CG/CL Hydrogel

XRD patterns of the kappa-carrageenan (κ-CG), cellulose (CL), and κ-CG/CL hydrogels are shown in Figure 2. As shown in Figure 2a, the XRD pattern of κ-CG with a broad hump in the range from 10° to 25° reveals its semi-crystalline nature. The (111), (002), and (310) planes were represented by three notable peaks at 2θ = 15.2°, 22.6°, and 34.4°, respectively [42]. The CL crystalline structure was revealed by the peak with the maximum angle at 2θ = 22.6° [43]. The diffractogram of the κ-CG/CL hydrogel exhibited a modest displacement and disappearance of the CL peaks and the XRD spectrum, as shown in Figure 2c. In this instance, the cross-linking of κ-CG with CL by shattering the semi-crystalline structure into an amorphous structure was confirmed by the small shift of the peaks at 2θ = 15.9° and 22.2° in the XRD pattern of the κ-CG/CL hydrogel. This reveals the active sites, enabling $Pb^{2+}$ ions to bind to the κ-CG/CL hydrogel.

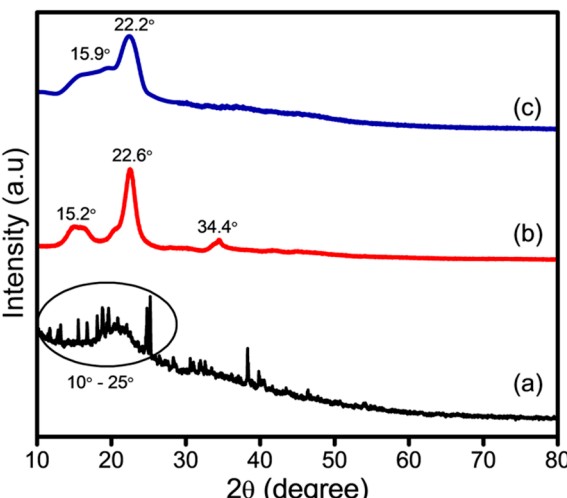

**Figure 2.** XRD pattern of (a) kappa-carrageenan (κ-CG), (b) cellulose (CL), and (c) κ-CG/CL hydrogels.

### 3.3. Morphology of κ-CG/CL Hydrogel

Figure 3 shows the SEM images of the κ-CG/CL hydrogel under different magnifications. The developed κ-CG/CL hydrogel is ideal for metal ion adsorption because of its heterogeneous surface and porous network structure with varying pore sizes, as observed for the κ-CG/CL hydrogel [44]. The hydrogel's high porosity allowed water to permeate and absorb easily. Figure 3 shows the largest and smallest pores in the κ-CG/CL hydrogel. Figure S1 shows a SEM image of the κ-CG/CL hydrogel with an open porous structure and

pore size of 1–10 μm. The pores of the κ-CG/CL hydrogel were found to be micro-sized. Therefore, electrostatic forces generated by the carboxylate anions (COO−) in the gel have expanded the space of the cross-linked κ-CG/CL hydrogel matrix. The hydrogel's high porosity would allow for rapid mass permeability, benefiting superabsorbent applications. Moreover, the κ-CG/CL hydrogel features characteristics that encourage swelling, which could aid the diffusion of heavy metal ions. In addition, nitrogen physisorption analysis of κ-CG/CL hydrogel shows a combination of type II adsorption–desorption isotherms and the plot of pore distribution as shown in Figure S2. The BET's large specific surface area, total pore volume, and average pore diameter were 419.3 m$^2$/g, 0.513 cm/g, and 269 nm, respectively.

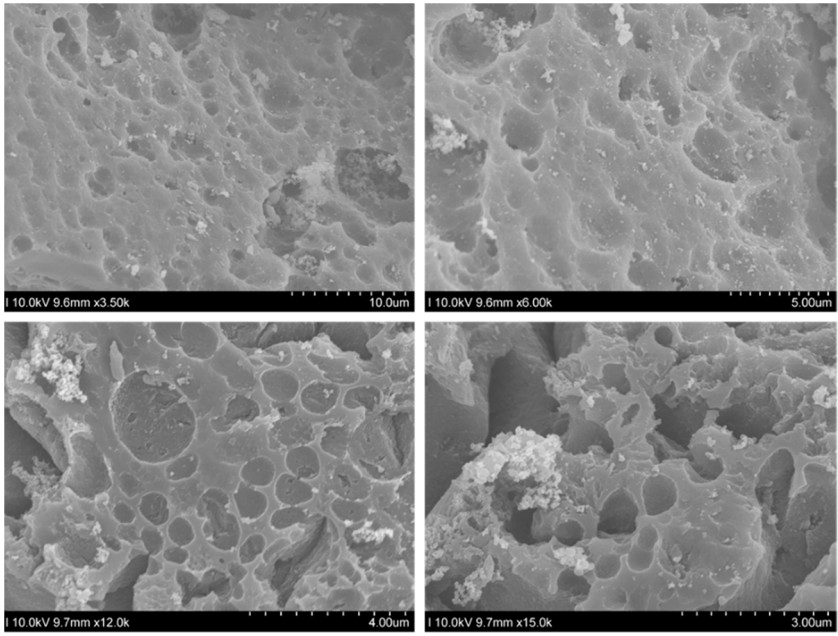

**Figure 3.** SEM images of κ-CG/CL hydrogel under different magnifications.

### 3.4. Adsorption Optimization of the κ-CG/CL Hydrogel

The adsorption capacity of Pb$^{2+}$ ions was studied as a function of pH. Metal hydroxides can precipitate from a solution when the pH is greater than 6, affecting the precision of the outcomes [45]. As the pH increases, the adsorption capacity for Pb$^{2+}$ ions increases, causing them to bind at high hydrogen-ion concentrations, as shown in Figure 4a. Meanwhile, the adsorbent surface was positively charged, and most major adsorption sites were protonated at low pH. Furthermore, this could result in a reduction of the stability and number of electrostatic interactions between the κ-CG/CS hydrogel and Pb$^{2+}$ ions [46]. A low pH resulted in reduced swelling of the adsorbent, decreasing the number of metal ions entering the κ-CG/CL hydrogel. A comprehensive assessment showed that pH 5.0 was the most suitable value [47]. Figure 4b illustrates the adsorption capacity of Pb$^{2+}$ ions by κ-CG/CS hydrogels at different contact times. The adsorption capacity of Pb$^{2+}$ ions gradually increased as the contact time increased after 5 min (Figure 4b) [48].

The k-CG/CL hydrogel has a three-dimensional loose and porous structure that allows Pb$^{2+}$ ions to get to the sites for adsorption [49] easily. In the absence of empty sites, the adsorbent was saturated with Pb$^{2+}$ ions in the binding sites. Additionally, the Pb$^{2+}$ ions reached adsorption equilibrium within 18 min, owing to a reduction in the adsorption rate. Figure 4c shows the influence of the primary amount of Pb$^{2+}$ ions adsorption on κ-CG/CL hydrogel. To determine the adsorption capacity after exposing the κ-CG/CL hydrogel (53–483 mg/L) to the Pb$^{2+}$ solution, the influence of adsorbent concentration on the weight of the κ-CG/CL hydrogel (i.e., 50–600 mg) was investigated (10 mL, pH 5.0). The maximum adsorption increased with increasing concentrations of Pb$^{2+}$ ions and reached a value of

$483 \pm 24.1$ mg/L [50]. As the concentration of $Pb^{2+}$ ions increases, the accumulation of $Pb^{2+}$ ions on the adsorbent also increases. A high adsorption capacity was achieved because the mass transfer rate increased due to increased ion-driving forces. Thereafter, with a further increase in initial concentration, the adsorption capacity remained unchanged or decreased, and the κ-CG/CS hydrogel for the high concentration of $Pb^{2+}$ ions solution decreased the specific surface area and adsorption sites.

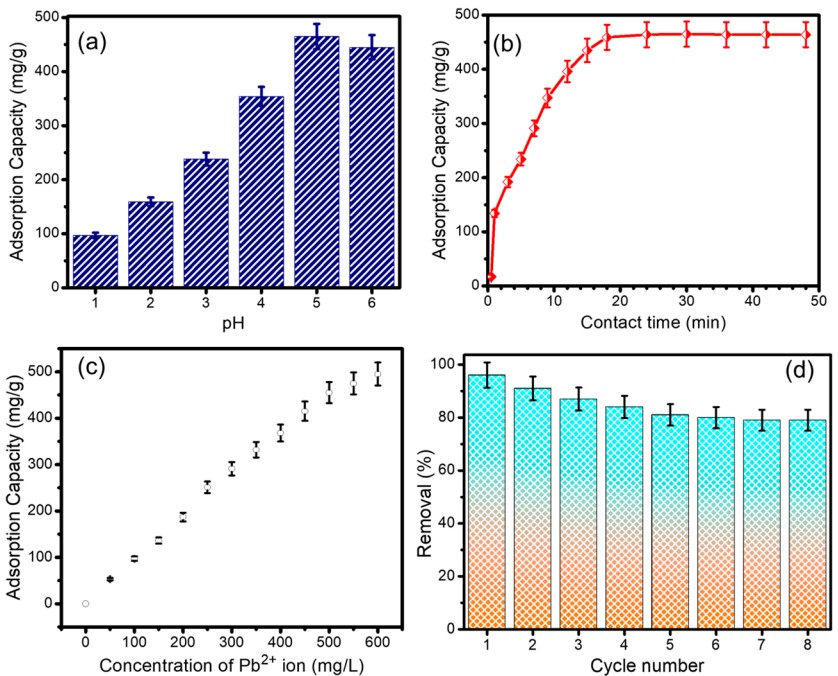

**Figure 4.** Effect of pH on (**a**) contact time, (**b**) $Pb^{2+}$ concentration, (**c**) adsorption capacity of the κ-CG/CL hydrogel, and (**d**) reusability after a cyclic run of κ-CG/CL hydrogel.

### *3.5. Reusability of κ-CG/CL Hydrogel*

A proper evaluation of the economy and application of the used adsorbent requires the recovery of metal ions from the adsorbing agent, as well as the regeneration of the used adsorbent. An overview of the reusability studies for $Pb^{2+}$ ions removal from κ-CG/CL hydrogel is presented in Scheme S1. The reusability experiment was conducted for eight subsequent cyclic runs. Figure 4d shows the percentages of the metal ions recovered from the κ-CG/CL hydrogel. In the first cycle, the removal efficiency was $96.1 \pm 4.8\%$, and by the end of the eighth cycle, the adsorption efficiency of $Pb^{2+}$ ions was $79.2 \pm 3.3\%$. Thus, even after eight cycles, the removal efficiency exceeded 79% [51]. These findings demonstrated that the κ-CG/CL hydrogel exhibited excellent reversibility and reusability, although some functional groups were partially removed after treatment. Because of its outstanding reusability, the produced κ-CG/CL hydrogel is a viable, affordable, and effective adsorbent to remove $Pb^{2+}$ ions.

### *3.6. Adsorption Kinetics of κ-CG/CL Hydrogel*

The most significant aspect of understanding adsorption is predicting the adsorption kinetics and their potential rate-limiting steps associated with mass transfer and reaction processes. To examine the controlling mechanism of the adsorption process, the most commonly used kinetic models, including the pseudo-first and second-order (PFO and PSO) (Equations (3) and (4)), were Elovich (E) (Equation (5)), and intra-particle diffusion (IPD) (Equation (6)) models to determine the underlying adsorption mechanism [52]. Figure 5 shows the kinetic fitting curve for $Pb^{2+}$ adsorption by the κ-CG/CL hydrogel, and Table S1 lists the relevant fitting parameters. A PFO kinetic model is commonly used to calculate liquid-phase adsorption kinetics, and in this model, the adsorption rate is

affected by diffusion and mass transfer. Using a PSO kinetic model, the reaction rate is linearly related to the two reactants' concentrations. In this case, it is assumed that chemical adsorption controls the rate of heavy metal ion adsorption. The adsorbent and adsorbate often share electrons during chemisorption. Figure 5a,b show the PFO and PSO fitting curves and parameters, respectively. Notably, the curves are nonlinear [53]. Table S1 shows that while the $Q_t$ values for the PSO model matched the experimental data closely, the computed $Q_t$ values for the PFO did not match the experimental results. The results of the PFO and PSO error functions of $Pb^{2+}$ were compared. The correlation coefficient for the PSO ($R^2 = 0.9959$) was greater than that for the PFO ($R^2 = 0.9863$) [54]. Additionally, it was discovered that the PFO represented the data more accurately than the PSO, which had higher chi-squared ($\chi^2$) values (402.27) [55]. These findings demonstrate that chemisorption predominates metal ion adsorption by the κ-CG/CL hydrogel and is consistent with the PSO equation.

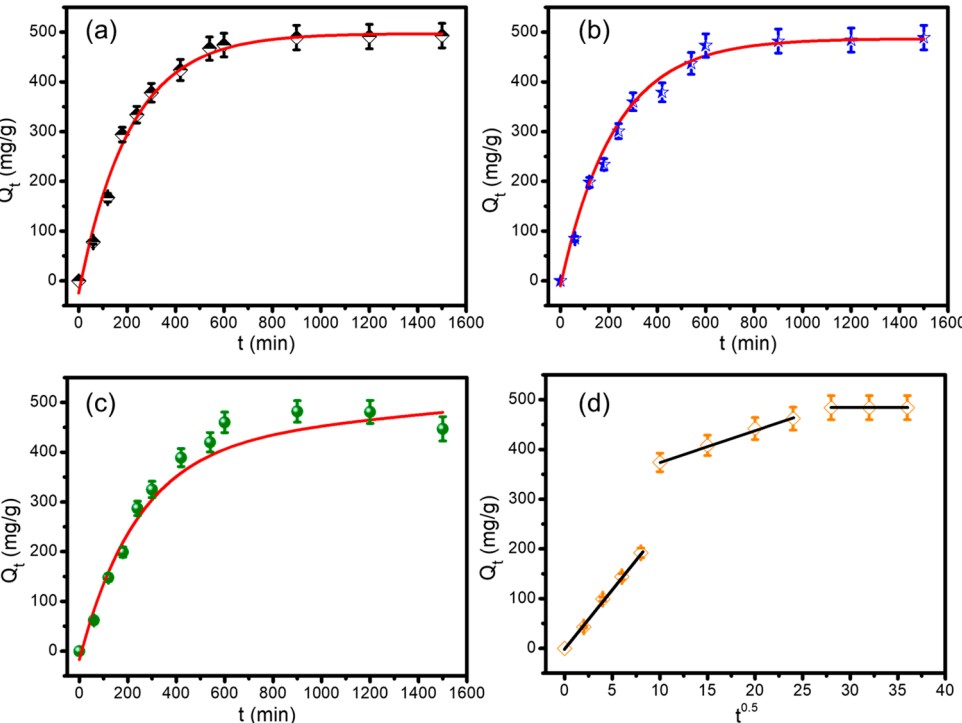

**Figure 5.** Fitting curves of the kinetics of $Pb^{2+}$ ion adsorption by κ-CG/CL hydrogel. (**a**) Pseudo-first-order, (**b**) pseudo-second-order, (**c**) Elovich, and (**d**) intra-particle diffusion models.

Furthermore, the PSO revealed a proportional correlation between the chemisorption rate and the square of the number of open adsorption sites. By the earlier analysis of the adsorption isotherm, the results were completely consistent. At primary concentrations of 50 and 500 mg/g, the PFO and PSO predict the adsorption outcomes more accurately than the E model (Figure 5c) [56]. Elovich (E) equations are empirical formulas that describe a process involving a series of reaction mechanisms. They include solutes in the solution phase, surface activation/deactivation, diffusion, etc. They are particularly useful for reactions in which the activation energy changes during the process. The E model may better fit the observed measurements at the primary concentration of 421.3 ± 19.3 mg/g because the adsorption process takes longer for large heavy-metal-ion doses. It is assumed that mass transfer processes and the effectiveness of the adsorption center have a significant impact on adsorption kinetics in the intraparticle diffusion (IPD) model. Figure 5d shows the three phases and multilinearity in the intra-particle diffusion plot obtained in this study. First, a considerably pronounced step resulted from $Pb^{2+}$ ion diffusion from the solution to the surface of the κ-CG/CL hydrogel and from the boundary layer to the surface.

Second, when the final equilibrium was reached, the low $Pb^{2+}$ ion concentration slowed intra-particle diffusion. Finally, the $Pb^{2+}$ ions in the solution slowed the intra-particle diffusion during the final equilibrium because of the low $Pb^{2+}$ ions content [57]. In general, diffusion rates decreased with increasing contact time because the $Pb^{2+}$ ions diffused into the inner structure of the κ-CG/CL hydrogel [58]. According to Table S1, the intercepts of the straight lines provide estimates of rate parameters, such as kinetics 1, 2, and 3, and the correlation coefficients. When the temperature and concentration of the $Pb^{2+}$ ions increased, the rate constants increased slightly. Therefore, diffusion and sorption occurred because of the higher concentration gradient. The correlation coefficient ($R^2$) associated with the PSO was much greater than those related to the PFO, E, and IPD models. Thus, the PSO describes the adsorption mechanism more accurately because it infers that the adsorption of $Pb^{2+}$ ions by the κ-CG/CL hydrogel is attributed to chemical adsorption [59].

### 3.7. Adsorption Isotherms of κ-CG/CL Hydrogel

The physicochemical adsorption of metal ions on adsorbent surfaces is described by adsorption isotherms, which illustrate the interfaces between the metal ions and the surfaces of the hydrogels [60]. Two major isotherm equations were used to determine the isotherm constants: the Langmuir and Freundlich equations. Based on the Langmuir isotherm, adsorption occurs on a homogeneous surface that has the same adsorption capacity for each adsorption site. Adsorption on a reversible heterogeneous surface is described by the Freundlich model based on interactions between adsorbed molecules [61]. The relationship between the initial $Pb^{2+}$ ion concentration and adsorption capacity in the process of $Pb^{2+}$ adsorption on κ-CG/CL hydrogels is shown in Figure 6a,b, and Table S2. The $Pb^{2+}$ adsorption capacity was low at low $Pb^{2+}$ concentrations and then increased slowly with an increase in the concentration of $Pb^{2+}$ [62]. The constant adsorption capacity indicates that the adsorption process has reached saturation. This phenomenon can be explained by two factors: (i) the gradual decrease in the mobility of $Pb^{2+}$ ions due to a gradual increase in the $Pb^{2+}$ ion concentration; (ii) the limited adsorbent dosage with inadequate binding sites [63]. As shown in Figure 6a,b, the relatively high $R^2$ values of the Langmuir and Freundlich isotherm models at 298, 308, and 318 K indicate that the models closely fit the $Pb^{2+}$ adsorption data of the κ-CG/CL hydrogel [64]. Particularly, the Langmuir isotherm model for $Pb^{2+}$ ions indicated high removal capacities of 454 ± 18.1, 418 ± 21.9, and 373 ± 11.4 mg/g at 298, 308, and 318 K, respectively. Similarly, in the case of the Freundlich isotherm model for $Pb^{2+}$ ions, the maximum removal capacities were 486 ± 28.5, 440 ± 23.3, and 388 ± 48.7 mg/g at three temperatures [65]. Consequently, the Freundlich isotherm model provided a more accurate description of adsorption at all three temperatures than the Langmuir isotherm model.

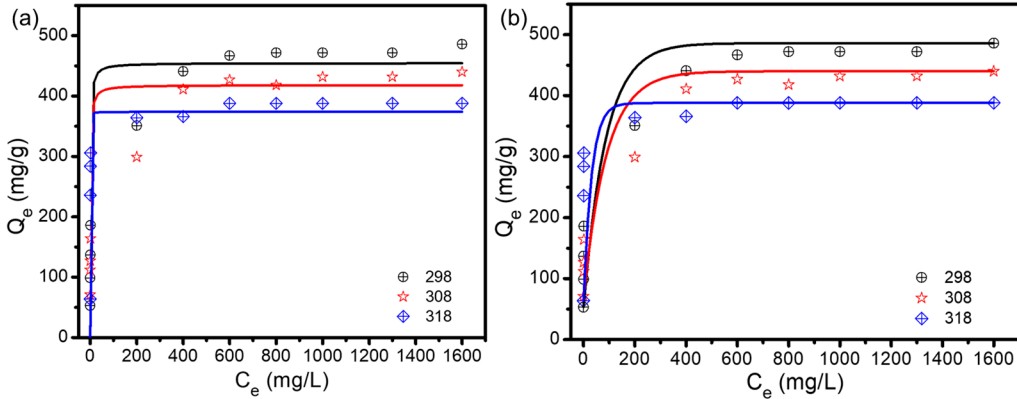

**Figure 6.** Fitting curves of $Pb^{2+}$ ion adsorption isotherms by κ-CG/CL hydrogel. (**a**) Langmuir and (**b**) Freundlich models.

The adsorbate and binding sites may interact differently, such as through ion exchange, complexation, electrostatic interactions, and chelation, and complicate the entire process [66]. These hydrogels exhibited a high affinity for $Pb^{2+}$ ions because of the large number of functional groups on their surfaces. The chelation of $Pb^{2+}$ ions with these functional groups is also possible [67]. The κ-CG/CL hydrogel adsorption mechanism was investigated. The peak of O−H stretching vibration was at 3315 $cm^{-1}$, and the band of a carboxyl group (−C=O) stretching vibration was at 1645 $cm^{-1}$, as seen in the FTIR spectra (Figure S3). Both bands moved to 3309 and 1677 $cm^{-1}$ after $Pb^{2+}$ ions adsorption, respectively. Meanwhile, two new peaks related to $Pb^{2+}$ ions species emerged at 923 and 841 $cm^{-1}$. A significant change in wavenumbers was found in the spectra of κ-CG/CL hydrogel before and after the adsorption of $Pb^{2+}$ ions. The -OH and -COOH groups are broken by $Pb^{2+}$ ions. After that, O provides an electron pair to the unoccupied orbital of $Pb^{2+}$ ionS, thereby forming O-Pb or CO-Pb interactions. The FTIR study results verified the adsorption of $Pb^{2+}$ ions on the functional groups of the κ-CG/CL hydrogel surface. To examine the adsorption kinetics of $Pb^{2+}$ ions on the κ-CG/CL hydrogel using adsorption kinetics and to explain the basic adsorption process, we fitted the experimental data to PFO, PSO, E, and IPD models. Moreover, the adsorption isotherm of the κ-CG/CL hydrogel also fits the Langmuir and Freundlich models well, suggesting the adsorption rate on the hydrogel surface [68].

### 3.8. Comparison with Other Adsorbents

The adsorption capacity of the κ-CG/CL hydrogel for $Pb^{2+}$ ions was compared with those of other adsorbents, as summarized in Table S3. The adsorption of heavy metal ions using other adsorbents has been widely reported. Based on the Freundlich isotherm model, the maximum adsorption capacity of the κ-CG/CL hydrogel is estimated to be $486 \pm 28.5$ mg/g. The data presented here support that our adsorbent has a higher maximum adsorption capacity than the previously reported adsorbents [69–72]. The results show that the active functional group -OH, which can obtain metal ions by exchange, simple chelation, or adsorption owing to the opening of the polymer matrix, provides the κ-CG/CL hydrogel with a high adsorption capacity. The κ-CG/CL hydrogel is evaluated in performance during the adsorption process using robust parameter optimization. A potential application of these new κ-CG/CL hydrogels is the removal of heavy metal ions from wastewater and aqueous effluents.

### 4. Conclusions

In summary, a κ-CG/CL hydrogel was successfully prepared using $CaCO_3$ to adsorb $Pb^{2+}$ ions from aqueous solutions. FTIR analysis of the hydrogel revealed that it comprises a physically cross-linked network. The XRD results showed numerous binding sites for heavy-metal ions in the structure of the κ-CG/CL hydrogel. The surface of the k-CG/CL hydrogel was porous. The BET's large specific surface area, total pore volume, and average pore diameter were 419.3 $m^2$/g, 0.513 cm/g, and 269 nm, respectively. This study examined the pH, contact time, and initial metal concentration as factors affecting adsorption. $Pb^{2+}$ ions must be adsorbed at pH 5.0 or higher to maximize the effectiveness of this material. The -OH groups attached to $Pb^{2+}$ ions, as indicated by the isotherms, and the adsorption kinetics resulted in a chelate compound. The adsorption process of κ-CG/CL hydrogel can be explained by pseudo-first/second-order kinetic, Elovich, and intra-particle diffusion models. Additionally, our kinetics study indicated that the adsorption process is regulated by two processes, namely surface diffusion and pore diffusion. The Langmuir isotherm model exhibited the best fit with the experimental data, and the $Pb^{2+}$ adsorption process was considerably effective. Based on the fitting results, the maximum adsorption capacity was obtained with the Freundlich isotherm model of κ-CG/CL hydrogel found to be $486 \pm 28.5$ mg/g[1]. Reusability studies revealed that the κ-CG/CL hydrogel could remove $Pb^{2+}$ ions with more than 79% efficiency after eight adsorption–desorption cycles. Therefore,

κ-CG/CL hydrogels are eco-friendly, efficient, and reusable adsorbents for removing heavy metals from aqueous solutions.

**Supplementary Materials:** The following supporting information can be downloaded at https://www.mdpi.com/article/10.3390/su15129534/s1. Figure S1: Pore size measurements of κ-CG/CL hydrogel. Figure S2: (a) BET analysis and (b) pore size distribution of the κ-CG/CL hydrogel. Figure S3: FTIR spectra of κ-CG/CL hydrogel before and after $Pb^{2+}$ ions adsorption. Scheme S1: An overview of the reusability studies for $Pb^{2+}$ ions removal from κ-CG/CL hydrogel. Table S1: Adsorption kinetics model parameters for $Pb^{2+}$ ions removal by κ-CG/CL hydrogel. Table S2: Adsorption isotherm model parameters for $Pb^{2+}$ ions removal by κ-CG/CL hydrogel. Table S3: Comparison of maximum $Pb^{2+}$ ions adsorption capacity of other reported adsorbents. References [73–82] are cited in the supplementary materials.

**Author Contributions:** K.K., methodology, formal analysis, writing—original draft; S.M., writing—review and editing; N.A., visualization, writing—review; M.R.K., methodology, formal analysis; R.K.M., supervision, conceptualization, writing—review and editing. All authors have read and agreed to the published version of the manuscript reported.

**Funding:** This research was funded by the Researchers Supporting Project Number (RSPD-2023R668), King Saud University, Riyadh, Saudi Arabia.

**Institutional Review Board Statement:** Not applicable.

**Informed Consent Statement:** Not applicable.

**Data Availability Statement:** All data generated or analyzed during this study are included in the published article.

**Acknowledgments:** The authors would also like to thank the Researchers Supporting Project Number (RSPD-2023R668), King Saud University, Riyadh, Saudi Arabia.

**Conflicts of Interest:** The authors declare no conflict of interest.

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
