# Peer review of "Adsorption of Pb2+ Ions from Aqueous Solution onto Porous Kappa-Carrageenan/Cellulose Hydrogels: Isotherm and Kinetics Study"

_sustainability, doi:10.3390/su15129534_

Round 1

Reviewer 1 Report

I suggest the following changes and improvements:

1.     Abstract: Please explain all abbreviations when they appear in the manuscript for the first time and then use the abbreviated form.

2.     The current structure of the introduction is not well organized. The authors need to be improved. Additionally, The novelty of this work should be stated clearly in the introduction section.

3.     What are the current research gap and research significance of this work?

4.     The author should provide a comparison-finding table of this study with other reported studies.

5.     The authors must be explaining their work's potential for use in practical applications.

Author Response

We thank Reviewer#1 for the favorable reception of our work and for highlighting the positive points in our study. We have revised our manuscript taking into great consideration all the comments and suggestions. Thank you for helping us to improve our manuscript.

Attached File: 

Reviewer #1:

I suggest the following changes and improvements:

We thank Reviewer#1 for the favorable reception of our work and for highlighting the positive points in our study. We have revised our manuscript taking into great consideration all the comments and suggestions. Thank you for helping us to improve our manuscript.

Comment 1

Abstract: Please explain all abbreviations when they appear in the manuscript for the first time and then use the abbreviated form.

Response 1

Thank you very much for your thoughtful and insightful suggestion. As suggested, we have explained all the abbreviations in this manuscript. Please see the revised manuscript.

Comment 2

The current structure of the introduction is not well organized. The authors need to be improved. Additionally, The novelty of this work should be stated clearly in the introduction section.

Response 2

Thank you very much for your thoughtful and insightful suggestion. As suggested, we have improved the introduction section, and also, we have provided the novelty of this work in this introduction section. Please see the revised manuscript.

Comment 3

What are the current research gap and research significance of this work?

Response 3

Thank you very much for your thoughtful and insightful suggestion. As suggested, we have explained the research significance of this work in the manuscript. Please see the revised manuscript.

Comment 4

The author should provide a comparison-finding table of this study with other reported studies.

Response 4

Thank you very much for your thoughtful and insightful suggestion. As suggested, we have provided the comparison finding table of this study with other reported adsorbents and metal ions. Please see the revised manuscript and supplementary materials.

Comment 5

The authors must be explaining their work's potential for use in practical applications.

Response 5

Thank you very much for your thoughtful and insightful suggestion. As suggested, we have explained the working potential for use in practical applications in this results and discussion section. Please see the revised manuscript.

Reviewer 2 Report

This paper describes the adsorption of Pb2+ ions on kappa-carrageenan/cellulose hydrogels. The topic of this research is important due to the need of treatment methods that can be used in the removal of toxic metal ions from aqueous solutions. The adsorbent is a sustainable material which can be easily prepared from two natural products. This new material has been characterized and its applicability to the removal of Pb2+ ions has been studied.

The work seems to be new, and the manuscript is generally well written and organized.

The results presented in Figure 7 are not clear to me. The two graphics (a) and (b) should represent the same experimental data, adjusted to two different models (Langmuir and Freundlich). However, the effect of the temperature is not the same on the two graphics, why? It should be expected that the adsorption would decrease with an increase in temperature.

Other specific comments are made below.

Line 67                 “An extract of a species of red seaweed called κ-CG…”

The definition of κ-CG can be improved.

Lines 93 – 95      “κ-CG/CL-based hydrogels were formulated using a 60: 40 κ-CG: CL ratio with 1.0% 93 CaCO3. For a typical hydrogel synthesis, 0.50 g of κ-CG and 0.20 g of CL…”

The ratio 60:40 seems not to be in agreement with the masses of reagents used (0.50:0.20). Please check.

Line 98                 “Before drying at 37 °C overnight in an oven at room temperature,…”               “Before drying at 37 °C overnight in an oven,…”

Line 112              300 RMP             300 RPM

Line 136              “…adsorption kinetics demonstration a significant part…”                         “…adsorption kinetics demonstration is a significant part…”

Line 165 – 170                  Regarding the adsorption isotherms, the equations presented by the authors (Eq. 7 and 8) do not correspond to the most traditional forms of the Langmuir and Freundlich equations. Please provide bibliographic references.

Line 190              Figure 1 caption should be improved. Please specify the compounds for (a), 8b) and (c)

Line 192              Please improve first sentence.

Line 204              Figure 2 caption. Please specify the compounds for (a), 8b) and (c)

Line 215              “…as a utility of pH.”                      “…as a function of pH.”

Line 242              “Because of the increased Pb2+ ion concentration in the solution, the specific surface area and adsorption sites of the κ-CG/CL hydrogel decreased.”

The specific surface area is a characteristic of the adsorbent that does not depend on the adsorbate concentration. The free adsorption sites decrease. Please rephrase.

Lines 304, 310, 320         Figure 7 should be renumbered Figure 6.

Line 309              “…absorption sites…”                    “…adsorption sites…”

Author Response

We thank Reviewer#2 for the favorable reception of our work and for highlighting the positive points in our study. We have revised our manuscript taking into great consideration all the comments and suggestions. Thank you for helping us to improve our manuscript.

Attached File: 

Reviewer #2:

This paper describes the adsorption of Pb2+ ions on kappa-carrageenan/cellulose hydrogels. The topic of this research is important due to the need of treatment methods that can be used in the removal of toxic metal ions from aqueous solutions. The adsorbent is a sustainable material which can be easily prepared from two natural products. This new material has been characterized and its applicability to the removal of Pb2+ ions has been studied. The work seems to be new, and the manuscript is generally well written and organized.

We thank Reviewer#2 for the favorable reception of our work and for highlighting the positive points in our study. We have revised our manuscript taking into great consideration all the comments and suggestions. Thank you for helping us to improve our manuscript.

Comment 1

The results presented in Figure 7 are not clear to me. The two graphics (a) and (b) should represent the same experimental data, adjusted to two different models (Langmuir and Freundlich). However, the effect of the temperature is not the same on the two graphics, why? It should be expected that the adsorption would decrease with an increase in temperature.

Response 1

Thank you very much for your thoughtful and insightful suggestion. As suggested, we have corrected the aforementioned issues in this manuscript. Please see the revised manuscript.

Comment 2

Line 67 “An extract of a species of red seaweed called κ-CG…”.

Response 2

Thank you very much for your thoughtful and insightful suggestion. As suggested, we have corrected the aforementioned sentence in this manuscript. Please see the revised manuscript.

Comment 3

The definition of κ-CG can be improved.

Response 3

Thank you very much for your thoughtful and insightful suggestion. As suggested, we have improved the definition of κ-CG in this manuscript. Please see the revised manuscript.

Comment 4

Lines 93 – 95 “κ-CG/CL-based hydrogels were formulated using a 60: 40 κ-CG: CL ratio with 1.0% 93 CaCO3. For a typical hydrogel synthesis, 0.50 g of κ-CG and 0.20 g of CL…”.

Response 4

Thank you very much for your thoughtful and insightful suggestion. As suggested, we have corrected the aforementioned sentence in this manuscript. Please see the revised manuscript.

Comment 5

The ratio 60:40 seems not to be in agreement with the masses of reagents used (0.50:0.20). Please check.

Response 5

Thank you very much for your thoughtful and insightful suggestion. As suggested, we have corrected the aforementioned correction in this manuscript. Please see the revised manuscript.

Comment 6

Line 98 “Before drying at 37 °C overnight in an oven at room temperature,…”               “Before drying at 37 °C overnight in an oven,…”.

Response 6

Thank you very much for your thoughtful and insightful suggestion. As suggested, we have corrected the aforementioned sentence in this manuscript. Please see the revised manuscript.

Comment 7

Line 112     300 RMP      300 RPM.

Response 7

Thank you very much for your thoughtful and insightful suggestion. As suggested, we have corrected the aforementioned mistakes in this manuscript. Please see the revised manuscript.

Comment 8

Line 136 “…adsorption kinetics demonstration a significant part…”                         “…adsorption kinetics demonstration is a significant part…”.

Response 8

Thank you very much for your thoughtful and insightful suggestion. As suggested, we have corrected the aforementioned mistakes in this manuscript. Please see the revised manuscript.

Comment 9

Line 165 – 170. Regarding the adsorption isotherms, the equations presented by the authors (Eq. 7 and 8) do not correspond to the most traditional forms of the Langmuir and Freundlich equations. Please provide bibliographic references.

Response 9

Thank you very much for your thoughtful and insightful suggestion. As suggested, we have cited the aforementioned adsorption isotherms equation references in materials and methods section. Please see the revised manuscript.

Comment 10

Line 190   Figure 1 caption should be improved. Please specify the compounds for (a), 8b) and (c).

Response 10

Thank you very much for your thoughtful and insightful suggestion. As suggested, we have specified the compounds of Figure 1. Please see the revised manuscript.

Comment 11

Line 192      Please improve first sentence…”.

Response 11

Thank you very much for your thoughtful and insightful suggestion. As suggested, we have improved the aforementioned sentence in this manuscript. Please see the revised manuscript.

Comment 12

Line 204    Figure 2 caption. Please specify the compounds for (a), 8b) and (c).

Response 12

Thank you very much for your thoughtful and insightful suggestion. As suggested, we have specified the compounds of Figure 2. Please see the revised manuscript.

Comment 13

Line 215       “…as a utility of pH.”               “…as a function of pH.”.

Response 13

Thank you very much for your thoughtful and insightful suggestion. As suggested, we have corrected the aforementioned mistakes in this manuscript. Please see the revised manuscript.

Comment 14

Line 242   “Because of the increased Pb2+ ion concentration in the solution, the specific surface area and adsorption sites of the κ-CG/CL hydrogel decreased.”.

Response 14

Thank you very much for your thoughtful and insightful suggestion. As suggested, we have improved the aforementioned sentence in this manuscript. Please see the revised manuscript.

Comment 15

The specific surface area is a characteristic of the adsorbent that does not depend on the adsorbate concentration. The free adsorption sites decrease. Please rephrase.

Response 15

Thank you very much for your thoughtful and insightful suggestion. As suggested, we have improved the aforementioned sentence in this manuscript. Please see the revised manuscript.

Comment 16

Lines 304, 310, 320    Figure 7 should be renumbered Figure 6..

Response 16

Thank you very much for your thoughtful and insightful suggestion. As suggested, we have corrected the aforementioned Figure number correction in this manuscript. Please see the revised manuscript.

Comment 17

Line 309         “…absorption sites…”             “…adsorption sites…”.

Response 17

Thank you very much for your thoughtful and insightful suggestion. As suggested, we have improved the aforementioned sentence in this manuscript. Please see the revised manuscript.

Reviewer 3 Report

The manuscript and the content are very good. I would suggest improving the final part of the abstract to highlight what is mentioned in the conclusions.

Author Response

We thank Reviewer#3 for the favorable reception of our work and for highlighting the positive points in our study. We have revised our manuscript taking into great consideration all the comments and suggestions. Thank you for helping us to improve our manuscript.

Attached File: 

Reviewer #3:

The manuscript and the content are very good. I would suggest improving the final part of the abstract to highlight what is mentioned in the conclusions.

We thank Reviewer#3 for the favorable reception of our work and for highlighting the positive points in our study. We have revised our manuscript taking into great consideration all the comments and suggestions. Thank you for helping us to improve our manuscript.

Comment 1

I would suggest improving the final part of the abstract to highlight what is mentioned in the conclusions.

Response 1

Thank you very much for your thoughtful and insightful suggestion. As suggested, we have improved the final part of the abstract section. Please see the revised manuscript.

Reviewer 4 Report

Please see the attached files for comments.

(Attached Files) 

Detailed comments:

In this paper, the authors developed a kappa carrageenan/cellulose hydrogel to effectively remove Pb2+ions from the aqueous solution. Their results indicate a high adsorption capacity of 489 mg/g and the authors analyzed the adsorption mechanisms by fitting the experimental data with different adsorption models. While the demonstrated performance is impressively high, the novelty need to be better articulated given that cellulose hydrogel has been studied long ago for heavy metal removal. For example, a similar work (Zhou, Yiming, et al. "ADSORPTION BEHAVIOR OF Cd 2+, Pb 2+, AND Ni 2+ FROM AQUEOUS SOLUTIONS ON CELLULOSE-BASED HYDROGELS." BioResources 7.3 (2012).) was published in 2012 that demonstrated ~ 400 mg/g adsorption capacity and analyzed with model fittings. Also, the overall data presentation and organization need improvement.

Major concerns:

1. The authors need to discuss about prior similar works and how this work is different from them in the introduction.

2. The demonstrated 5 cycles are not enough to show reusability. Please check the testing standard and increase cycle numbers.

3. Also, figure 4d shows a clear decreasing trend, which indicates the accumulation of Pd residue during usage. The authors need to suggest and a mitigation plan.

4. Section 3.6, please provide a brief introduction of each model about their assumption, strength, and limitation.

5. Section 3.8, organize and combine it with 3.7, discuss the motivation of modeling and conclude with more detailed discussion of mechanism.

6. Can authors test and discuss any potential applications for other heavy metal removal? E.g. Cd and Ni.

7. Line 71-72, the reduced mechanical properties seem to be adverse to the application. What’s the purpose of this statement?

8. Line 138-139, confusing statement, please clarify “equilibrium point in less than 1 min.

9. Add references to all the models used in section 2.6.

10. Figure 1 and 2 have no labels in the caption and the bonding and their wavenumber are not labeled in the figures. This makes it very difficult to follow the discussion. Please revise.

11. Line 223-226, please discuss the underlying reason for the observed trends.

12. “time” axis in figure 4b is confusing when the readers look at figure 4b and figure 5 together. Consider name figure 4b with “contact time”.

13. Many typos and wrong word choices, see minor concerns below.

14. Line 235-239, double check all the data and units. They seem to be misplaced.

15. Line 241, where is the result supporting this statement?

16. Line 242-243 doesn't make sense. Specific surface area and active sites are determined and controlled by the material not its operating condition.

17. Line 257, state explicitly what data is used for fitting.

Minor concerns:

1. line 50, wrong word choice “valuable”

2. line 55, grammar issue of “hydrogel exposed…”

3. line 57-60, wordy and repeating phrases, consider making it more concise.

4. Line 67, add transition between the sentences.

5. line 125, seems to be a typo of “volume of the Pb2+ ions”

6. Line 136, grammar issue, use verb “demonstrate”

7. Line 163, add reference.

8. Line 173, typo in the unit.

9. Line 178, 192 incomplete sentences.

10. Line 220, wrong word choice “method”

11. Line 246, wrong logic

12. Line 267, wrong transition “In addition”.

Please see the attached files for comments.

Author Response

We thank Reviewer#4 for the favorable reception of our work and for highlighting the positive points in our study. We have revised our manuscript taking into great consideration all the comments and suggestions. Thank you for helping us to improve our manuscript.

Attached File: 

Reviewer #4:

In this paper, the authors developed a kappa carrageenan/cellulose hydrogel to effectively remove Pb2+ ions from the aqueous solution. Their results indicate a high adsorption capacity of 489 mg/g and the authors analyzed the adsorption mechanisms by fitting the experimental data with different adsorption models. While the demonstrated performance is impressively high, the novelty need to be better articulated given that cellulose hydrogel has been studied long ago for heavy metal removal. For example, a similar work (Zhou, Yiming, et al. "ADSORPTION BEHAVIOR OF Cd2+, Pb2+, AND Ni2+ FROM AQUEOUS SOLUTIONS ON CELLULOSE-BASED HYDROGELS." BioResources 7.3 (2012).) was published in 2012 that demonstrated ~ 400 mg/g adsorption capacity and analyzed with model fittings. Also, the overall data presentation and organization need improvement.

We thank Reviewer#4 for the favorable reception of our work and for highlighting the positive points in our study. We have revised our manuscript taking into great consideration all the comments and suggestions. Thank you for helping us to improve our manuscript.

Major concerns:

Comment 1

The authors need to discuss about prior similar works and how this work is different from them in the introduction.

Response 1

Thank you very much for your thoughtful and insightful suggestion. As suggested, we have discussed this work compared with other reported work in the introduction section. Please see the revised manuscript.

Comment 2

The demonstrated 5 cycles are not enough to show reusability. Please check the testing standard and increase cycle numbers.

Response 2

Thank you very much for your thoughtful and insightful suggestion. As suggested, we have increased the cycle numbers for reusability studies. Please see the revised manuscript.

Comment 3

Also, figure 4d shows a clear decreasing trend, which indicates the accumulation of Pd residue during usage. The authors need to suggest and a mitigation plan.

Response 3

Thank you very much for your thoughtful and insightful suggestion. As suggested, we have corrected the aforementioned issue in the reusability studies. Please see the revised manuscript.

Comment 4

Section 3.6, please provide a brief introduction of each model about their assumption, strength, and limitation.

Response 4

Thank you very much for your thoughtful and insightful suggestion. As suggested, we have provided the brief introduction of each model characteristics of adsorption kinetics. Please see the revised manuscript.

Comment 5

Section 3.8, organize and combine it with 3.7, discuss the motivation of modeling and conclude with more detailed discussion of mechanism.

Response 5

Thank you very much for your thoughtful and insightful suggestion. As suggested, we have combined the sections 3.7 & 3.8 and the detailed adsorption mechanism discussed in this manuscript. Please see revised manuscript.

Comment 6

Can authors test and discuss any potential applications for other heavy metal removal? E.g. Cd and Ni”.

Response 6

Thank you very much for your thoughtful and insightful suggestion. I apologize for not being able to provide your valuable suggestions at this time. I can do in future as per your suggestion. I hope the reviewer reconsider this comment. 

Comment 7

Line 71-72, the reduced mechanical properties seem to be adverse to the application. What’s the purpose of this statement?

Response 7

Thank you very much for your comment. We have removed the unnecessary statement. Please see the revised manuscript.

Comment 8

Line 138-139, confusing statement, please clarify “equilibrium point in less than 1 min.

Response 8

Thank you very much for your thoughtful and insightful suggestion. As suggested, we have corrected the aforementioned sentences. Please see the revised manuscript.

Comment 9

Add references to all the models used in section 2.6.

Response 9

Thank you very much for your thoughtful and insightful suggestion. As suggested, we have added references in all the kinetic models. Please see the revised manuscript.

Comment 10

Figure 1 and 2 have no labels in the caption and the bonding and their wavenumber are not labeled in the figures. This makes it very difficult to follow the discussion. Please revise.

Response 10

Thank you very much for your thoughtful and insightful suggestion. As suggested, we have provided the wavenumber/2θ values and labels in the Figures 1 and 2. Please see the revised manuscript.

Comment 11

Line 223-226, please discuss the underlying reason for the observed trends.

Response 11

Thank you very much for your thoughtful and insightful suggestion. As suggested, we have discussed the aforementioned sentences. Please see the revised manuscript.

Comment 12

“time” axis in figure 4b is confusing when the readers look at figure 4b and figure 5 together. Consider name figure 4b with “contact time”.

Response 12

Thank you very much for your thoughtful and insightful suggestion. As suggested, we have changed the aforementioned mistake in Figure 4b. Please see the revised manuscript.

Comment 13

Many typos and wrong word choices, see minor concerns below.

Response 13

Thank you very much for your thoughtful and insightful suggestion. As suggested, we have corrected the typographical errors, word choices and superfluous spaces throughout the manuscript. Please see below or in the revised manuscript.

Comment 14

Line 235-239, double check all the data and units. They seem to be misplaced.

Response 14

Thank you very much for your thoughtful and insightful suggestion. As suggested, we have corrected the all the data units throughout the manuscript. Please see the revised manuscript and supplementary materials.

Comment 15

Line 241, where is the result supporting this statement?

Response 15

Thank you very much for your thoughtful and insightful suggestion. As suggested, we have removed the unnecessary sentences in this manuscript. Please see the revised manuscript.

Comment 16

Line 242-243 doesn't make sense. Specific surface area and active sites are determined and controlled by the material not its operating condition.

Response 16

Thank you very much for your thoughtful and insightful suggestion. As suggested, we have corrected the aforementioned sentence. Please see the revised manuscript.

Comment 17

Line 257, state explicitly what data is used for fitting.

Response 17

Thank you very much for your thoughtful and insightful suggestion. As suggested, we have corrected the aforementioned issue in this manuscript. Please see the revised manuscript.

Minor concerns:

Comment 1

line 50, wrong word choice “valuable”.

Response 1

Thank you very much for your thoughtful and insightful suggestion. As suggested, we have corrected the aforementioned sentences in the introduction section. Please see the revised manuscript.

Comment 2

line 55, grammar issue of “hydrogel exposed…”.

Response 2

Thank you very much for your thoughtful and insightful suggestion. As suggested, we have corrected the aforementioned sentences in the introduction section. Please see the revised manuscript.

Comment 3

line 57-60, wordy and repeating phrases, consider making it more concise.

Response 3

Thank you very much for your thoughtful and insightful suggestion. As suggested, we have corrected and concise the aforementioned sentences in the introduction section. Please see the revised manuscript.

Comment 4

Line 67, add transition between the sentences..

Response 4

Thank you very much for your thoughtful and insightful suggestion. As suggested, we have corrected the aforementioned sentences in the introduction section. Please see the revised manuscript.

Comment 5

line 125, seems to be a typo of “volume of the Pb2+ ions”.

Response 5

Thank you very much for your thoughtful and insightful suggestion. As suggested, we have corrected the aforementioned mistakes in the Materials and methods section. Please see the revised manuscript.

Comment 6

Line 136, grammar issue, use verb “demonstrate”.

Response 6

Thank you very much for your thoughtful and insightful suggestion. As suggested, we have corrected the aforementioned grammar issue in the Materials and methods section. Please see the revised manuscript.

Comment 7

Line 163, add reference.

Response 7

Thank you very much for your thoughtful and insightful suggestion. As suggested, we have cited the aforementioned adsorption isotherms equation references in materials and methods section. Please see the revised manuscript.

Comment 8

Line 173, typo in the unit.

Response 8

Thank you very much for your thoughtful and insightful suggestion. As suggested, we have corrected the aforementioned typo mistakes in the Materials and methods section. Please see the revised manuscript.

Comment 9

Line 178, 192 incomplete sentences..

Response 9

Thank you very much for your thoughtful and insightful suggestion. As suggested, we have corrected the aforementioned sentences in the introduction section. Please see the revised manuscript.

Comment 10

Line 220, wrong word choice “method”.

Response 10

Thank you very much for your thoughtful and insightful suggestion. As suggested, we have corrected the aforementioned sentences in the introduction section. Please see the revised manuscript.

Comment 11

Line 246, wrong logic.

Response 11

Thank you very much for your thoughtful and insightful suggestion. As suggested, we have corrected the aforementioned sentences in the introduction section. Please see the revised manuscript.

Comment 12

Line 267, wrong transition “In addition”.

Response 12

Thank you very much for your thoughtful and insightful suggestion. As suggested, we have corrected the aforementioned sentences in the introduction section. Please see the revised manuscript.

Reviewer 5 Report

Dear Author(s),

In their study, the author(s) investigated (I) the efficiency of a κ-CG/CL hydrogel they developed in removing Pb+2 ions from aqueous solutions, (II) the effects of pH, dosage, contact time, concentration and regeneration variables on the adsorption of the Pb+2 ion from aqueous solution, and (III) the Langmuir and Freundlich isotherms and the kinetics and adsorption mechanism of Pb2+ ions. The paper is well written, has current and important data, and should be of great interest to the readers. The introduction and others sections provide useful information for the readers. Some points have to be clarified or fixed. 

I here summarize this points:

1- In addition to the maximum adsorption capacity (489.3 ± 23.9 mg.g-1) in the abstract, it will be more remarkable if the percentage removal efficiency is given in parentheses [for instance 489.3 ± 23.9 mg.g-1 (81% removal efficiency)].

2- Some criteria are taken into account when choosing an adsorbent. Some of these are being environmentally friendly, easy to make/supply, economical, etc. In this study, it should be explained why "kappa-carrageenan/cellulose hydrogels" was chosen as an adsorbent.

3- In the introduction of the text, the effects of heavy metals on the food chain (Line 31) are stated in a single sentence. This part should be detailed by giving examples of the presence of heavy metals in foods (seafoods, milk, fruits and vegetables, etc.). You can take advantage of the following articles.

Seafoods

- Yabanlı, M., Åžener, İ., Yozukmaz, A., Öner, S., & Yapıcı, H. H. (2021). Heavy metals in processed seafood products from Turkey: risk assessment for the consumers. Environmental Science and Pollution Research, 28(38), 53171-53180.

Milk

- Zhou, X., Zheng, N., Su, C., Wang, J., & Soyeurt, H. (2019). Relationships between Pb, As, Cr, and Cd in individual cows’ milk and milk composition and heavy metal contents in water, silage, and soil. Environmental Pollution, 255, 113322.

Fruits and vegetables

Shaheen, N., Irfan, N. M., Khan, I. N., Islam, S., Islam, M. S., & Ahmed, M. K. (2016). Presence of heavy metals in fruits and vegetables: Health risk implications in Bangladesh. Chemosphere, 152, 431-438.

4- Line 87: The sentence should start with a capital letter (Kappa-).

5- Information should be provided on the calibration solution used to determine Pb2+ residue concentrations using AAS.

6- In the materials and methods section, statistical analysis should be included as a subheading.

The content of the article in accordance with the aims of the Sustainability.

The article is scientifically sufficient.

The article is scientifically sufficient.

Keywords are well chosen so that the article can be found by indexes.

The literature has been adequately critical, current and internationally evaluated by the authors.

The language of the article is correct and clear.

The discussion part is quite comprehensive in the paper.

Tables and figures are well designed and necessary.

Author Response

We thank Reviewer#5 for the favorable reception of our work and for highlighting the positive points in our study. We have revised our manuscript taking into great consideration all the comments and suggestions. Thank you for helping us to improve our manuscript.

Attached File: 

Reviewer #5:

In their study, the author(s) investigated (I) the efficiency of a κ-CG/CL hydrogel they developed in removing Pb2+ ions from aqueous solutions, (II) the effects of pH, dosage, contact time, concentration and regeneration variables on the adsorption of the Pb2+ ion from aqueous solution, and (III) the Langmuir and Freundlich isotherms and the kinetics and adsorption mechanism of Pb2+ ions. The paper is well written, has current and important data, and should be of great interest to the readers. The introduction and others sections provide useful information for the readers. Some points have to be clarified or fixed.

We thank Reviewer#5 for the favorable reception of our work and for highlighting the positive points in our study. We have revised our manuscript taking into great consideration all the comments and suggestions. Thank you for helping us to improve our manuscript.

Comment 1

In addition to the maximum adsorption capacity (489.3 ± 23.9 mg.g-1) in the abstract, it will be more remarkable if the percentage removal efficiency is given in parentheses [for instance 489.3 ± 23.9 mg.g-1 (81% removal efficiency)].

Response 1

Thank you very much for your thoughtful and insightful suggestion. As suggested, we have corrected the aforementioned sentence in this manuscript. Please see the revised manuscript.

Comment 2

Some criteria are taken into account when choosing an adsorbent. Some of these are being environmentally friendly, easy to make/supply, economical, etc. In this study, it should be explained why "kappa-carrageenan/cellulose hydrogels" was chosen as an adsorbent.

Response 2

Thank you very much for your thoughtful and insightful suggestion. As suggested, we have explained the details of choosing an adsorbent in this work. Please see the revised manuscript.

Comment 3

In the introduction of the text, the effects of heavy metals on the food chain (Line 31) are stated in a single sentence. This part should be detailed by giving examples of the presence of heavy metals in foods (seafoods, milk, fruits and vegetables, etc.). You can take advantage of the following articles.

  • Seafoods - Yabanlı, M., Åžener, İ., Yozukmaz, A., Öner, S., & Yapıcı, H. H. (2021). Heavy metals in processed seafood products from Turkey: risk assessment for the consumers. Environmental Science and Pollution Research, 28(38), 53171-53180.
  • Milk - Zhou, X., Zheng, N., Su, C., Wang, J., & Soyeurt, H. (2019). Relationships between Pb, As, Cr, and Cd in individual cows’ milk and milk composition and heavy metal contents in water, silage, and soil. Environmental Pollution, 255, 113322.
  • Fruits and vegetables - Shaheen, N., Irfan, N. M., Khan, I. N., Islam, S., Islam, M. S., & Ahmed, M. K. (2016). Presence of heavy metals in fruits and vegetables: Health risk implications in Bangladesh. Chemosphere, 152, 431-438.

Response 3

Thank you very much for your thoughtful and insightful suggestion. As suggested, we have cited the aforementioned references in the introduction section. Please see the revised manuscript.

Comment 4

Line 87: The sentence should start with a capital letter (Kappa-).

Response 4

Thank you very much for your thoughtful and insightful suggestion. As suggested, we have corrected the aforementioned sentence in the 2.1. Materials section. Please see the revised manuscript.

Comment 5

Information should be provided on the calibration solution used to determine Pb2+ residue concentrations using AAS.

Response 5

Thank you very much for your thoughtful and insightful suggestion. As suggested, we have provided the calibration solution used to determine Pb2+ residue concentrations using AAS. Please see the revised manuscript and supplementary files.

Comment 6

In the materials and methods section, statistical analysis should be included as a subheading.

The content of the article in accordance with the aims of the Sustainability. The article is scientifically sufficient. The article is scientifically sufficient. Keywords are well chosen so that the article can be found by indexes. The literature has been adequately critical, current and internationally evaluated by the authors. The language of the article is correct and clear. The discussion part is quite comprehensive in the paper. Tables and figures are well designed and necessary.

Response 6

Thank you very much for your thoughtful and insightful suggestion. As suggested, we have provided the statistical analysis in the Materials and methods section. Please see the revised manuscript. Finally, we thank Reviewer for the favorable reception of our work and for highlighting the positive points in our study.

Reviewer 6 Report

My review for this manuscript sustainability-2362893

1. The absence of pore structure data in characterization of porous kappa-carrageenan/cellulose hydrogels, please supplement it.

2. Lack of mechanism and structure-performance analysis in the Abstract part, please improve it.

3. Author should add some recent literature for lignin based adsorbents in Introduction part

[1] Chemosphere, 2022, 288: 132499.

[2] International Journal of Biological Macromolecules, 2022, 221: 25-37.

[3] ACS Applied Polymer Materials, 2021, 3(4): 2178-2188.

4. Where is the innovation of this paper? I seem to don't see it. The introduction needs to be modified to enhance its novelty. The author described the existing Hydrogels are also synthesized using κ-CG and CL [20-22]. So what are the characteristics of the materials you work with?

5. Page 3, line 98, Before drying at 37 °C overnight in an oven at room temperature, so which is this room temperature or 37 °C after all?

6.  For this sentence, the adsorption capacity decreased or remained constant with increasing initial concentrations.

There is something wrong with this description. There is no decline, only the adsorption saturation can be reached, and the adsorption amount is basically constant.

7. The part about adsorption mechanism should have some data characterization to provide strong proof.

8. Is the maximum adsorption capacity of adsorbent measured under the same condition in Table S3? In addition, the references in the supporting materials are not in the right format.

no

Author Response

We thank Reviewer#6 for the favorable reception of our work and for highlighting the positive points in our study. We have revised our manuscript taking into great consideration all the comments and suggestions. Thank you for helping us to improve our manuscript.

Round 2

Reviewer 6 Report

Now, author's response had not  solved all the problems from my suggestion. For example, question 1 and 8, especially, for question 1, the pore structure data can not be simply obtained  by SEM analysis, my suggestion was that please supplemented the pore structure including sepcific surface areas, pore volume, and pore size and distribution of materials by the N2 adsorption-desorption test ( BET characterization), which is very important structural parameters for adsorption performance.

good

Author Response

(The authors gave the same response as above.)
